# The Relationship between Children's Trait Emotional Intelligence and the Big Five, Big Two and Big One Personality Traits

Èlia López-Cassà [1,*] , Núria Pérez-Escoda [2] and Alberto Alegre [3]

1   Department of Didactics and Educational Organization, University of Barcelona, 08007 Barcelona, Spain
2   Department of Research Methods and Diagnosis in Education, University of Barcelona, 08007 Barcelona, Spain; nperezescoda@ub.edu
3   Department of Early Childhood and Elementary Education, East Stroudsburg University, East Stroudsburg, PA 18301, USA; merchalbert@yahoo.es
*   Correspondence: elialopez@ub.edu

**Abstract:** The irrefutable repercussions of personality and socio-emotional development on children's learning and psychological well-being justify the relevance for the educational context of delving into the relationship between those two constructs. Therefore, the research presented in this article investigates the link between trait EI and the B5, B2, and B1 (or GFP) personality traits in children between 9 and 13 years of age. We used the Spanish adaptation of the BFQ-NA (Big Five Personality Questionnaire for Children and Adolescents) and the CDE_9-13 (Emotional Development Questionnaire for primary education) with a sample of 259 primary school students. The results showed correlations between the two Big personality factors (B2) and the Big One personality factor (B1) with trait EI. However, the relationship between trait emotional intelligence and the Big Five personality model (B5) was not very high; only two of the five personality traits significantly predicted trait EI. Thus, our results differ from studies conducted with adults, but instead, it is similar to studies conducted with children. Finally, this study reinforces the thesis that trait EI can be considered a synonym of the GFP (General Factor Personality). Consequently, it implies designing and implementing learning and socioemotional development programs during the school years to promote adaptability and social efficacy.

**Keywords:** emotional intelligence; personality; children; personality traits; primary education

## 1. Introduction

Societies of the 21st century face significant challenges, among which is the improvement of education to provide people with the knowledge, skills, and tools they need throughout their lives [1]. For this reason, we are witnessing profound educational changes aimed at comprehensive development in skills, educational quality, responding to social demands, and giving visibility to emotional education as a subject in compulsory education.

Recently, the OECD [2] has presented an international report stating that the benefits of developing socio-emotional skills in children go beyond cognitive development and academic results; additionally, they are essential drivers of health and labor market prospects [2].

In this sense, programs designed to develop emotional competencies have shown positive results [3–5]. Numerous studies confirm that educational programs that include emotional education positively impact attitudes towards oneself and others, improve classroom climate, assertiveness, resilience, reduce disruptive behavior and increase academic performance, and other relevant aspects of quality education [3,6–8]. Moreover, they contribute to improving psychological well-being [9–11] while decreasing anxiety rates [6,12] and mental health problems [13].

Hence, several authors have defended that emotional education is considered as a primordial aspect of the integral development and well-being of students [14–17]. Emotional education is one of the life skills that should be emphasized, as it aims to provide students with a series of necessary tools that enable them to deal effectively and efficiently with the challenges, tasks, and situations that occur in their lives [17].

Emotional intelligence, therefore, plays a vital role in achieving success in most human activities [18], including those related to education [19]. In addition to emotional intelligence, personality plays a fundamental role in human behavior. Childhood personality is a proven predictor of many essential outcomes in adulthood [20,21]. Personality can be described as a set of stable characteristics and tendencies predisposing the person to behave in a certain way in various life situations [22].

The debate on these constructs has continued throughout the 21st century, where various theoretical frameworks have been developed on personality and similarly on emotional intelligence [23].

Therefore, knowing and analyzing the influence of Emotional Intelligence (EI) on children's personalities and how these variables affect their education is essential.

The study of personality has been conceptualized by two models, the Giant Tree model [24] and the Big Five model of personality [25]. Both taxonomies postulate that major personality traits are determinants of an individual's behavior, although the Big Five is the most widely accepted model of personality in the different fields of study. This model holds that we can study personality through five major factors: Openness, Responsiveness, Extraversion, Agreeableness, and Neuroticism that account for most individual differences [26]. However, following the work of Digman [27] and De Young et al. [28], these traits were grouped into two more prominent personality factors (B2): Alpha/Stability and Beta/Plasticity. The former would include agreeableness, responsibility, and neuroticism, while the latter would encompass openness and extroversion. This grouping has been used in several studies [29,30] and the same model has been applied across cultures, life stages (children, adolescents, and adults), and languages [31,32].

In contrast, other studies have revealed the existence of a general factor of personality (GFP) which is based on the idea that it is a dominant higher-order factor (Big one, B1) that binds and integrates all other personality components [33–35]. According to this theory, GFP would represent an evolutionarily adaptive trait with an inherited basis, although influenced by upbringing, socialization, and education [35–37].

Research on the five factors and personality, in general, has focused mainly on individual differences in adulthood but has not been as thoroughly examined within younger populations. However, an interesting research study conducted by Tackett et al. [21] provides the first examination of the hierarchical structure of childhood personality in five different countries and ages. Their results showed that the five-factor model was notable from early childhood, but as children grow, they experience changes in each personality dimension. Those deviations continue until adulthood. Thus, the need to study the development of children's personality traits has been increasingly recognized to understand how traits develop and change across the lifespan [38–40].

Emotional intelligence (EI) is a construct that first appeared in an article published by Salovey and Mayer [41]. Since then, a wide variety of models of emotional intelligence have proliferated, which can be grouped into two broad categories. One refers to the ability model, and the other to the trait model. The difference between the two tends to be clearly delineated as they are seen as distinct constructs [6].

The ability model considers EI as the ability to process emotional information [42], while Trait emotional intelligence (trait EI) relates to stable personality traits [43]. These authors defined EI as: "A set of emotional components and self-perceived abilities representing the personality construct" [43] (p. 479). The construct formulated by Petrides provides a clear operationalization of the emotion-related aspects of personality and lies outside the taxonomy of cognitive abilities and within the taxonomy of personality [44].

Petrides et al. [45] provided a comprehensive overview of the fields of application of trait EI. Although much research remains to be done, there is evidence of the importance of Trait EI in adults [46], children [47,48], and adolescents [49,50]. Research on Trait EI in children, the stage we will address in this paper, indicates that Trait EI is a significant predictor of health, well-being, social relationships, academic performance, and adaptive behaviors [51–53].

Some research argues that trait EI can be considered a lower-order personality trait [54,55], whereas others suggest that trait EI would be an indicator of the General Personality Factor (GPF) [56,57].

Meta-analysis studies have shown relatively high associations between GPF and trait EI [35,58–60] and have provided interesting findings confirming that all five personality traits (B5) correlate with trait EI in adult samples [58,60]. However, the GFP-trait EI association in children and adolescents appears to differ from that in adults due to the structures and development of personality during childhood [61]. Furthermore, the lower-order structure of youth traits appears to shift with age [40,62]. Recent research points out that several personality traits show distinctive developmental trends in children and youths. The mean levels of extraversion, agreeableness, conscientiousness, emotional stability, and openness decrease from late childhood to middle adolescence. Therefore, there is a decrease in personality maturity [61,63,64].

Given the above, we need to clarify these contributions and probe deeply into the relationship between trait EI and personality. This study uses a sample of Spanish children aged 9 to 13 years and a new measure of trait EI developed by Pérez-Escoda et al. [65].

Specifically, our research has three objectives:

1. To analyze the relationship between trait emotional intelligence and the Big Five (B5) model of personality.
2. To assess the correlation between trait EI and the two Big Five (B2) factors.
3. To test the relationship between trait EI and the Big One (B1) personality factor.

## 2. Materials and Methods

### 2.1. Procedures

The students answered the different questionnaires online with the presence and support of their teacher during a class session. The duration of the tests was approximately 45 min.

All students participated voluntarily, and the schools collected the corresponding consents signed by the children's families.

### 2.2. Participants

A total of 259 Spanish elementary students (49.4% female) completed the sample for this study. The participants ranged in age from 9 to 13 years old (M = 9.98, SD = 2.44) and in their level of studies from fourth to sixth grade of primary education.

### 2.3. Ethics Statement

This study was conducted following the recommendations of the Bioethics Committee of the University of Barcelona and subject to the ethical standards established by the 1964 Declaration of Helsinki and its subsequent modifications.

The research group signed a research agreement with each school guaranteeing the confidentiality of the results.

### 2.4. Measurement Instruments

*Personality*. The BFQ- NA (Big Five Personality Questionnaire for Children and Adolescents) [66], in its Spanish adaptation by Del Barrio et al. [67], was applied to measure the Big Five personality model in children. It consists of 65 items arranged on a five-point Likert-type scale, with values between one point "*Completely false for me*" and five points "*Completely true for me*". Soto et al. [40] reported the psychometric characteristics of the

BFQ-NA, obtaining a Cronbach's alpha between 0.78 and 0.88 while confirming the five-factor structure corresponding to the Big Five through confirmatory and exploratory factor analyses. In the present study, the Cronbach's alpha of the subscales was Conscientiousness $\alpha$ = 0.85, Agreeableness $\alpha$ = 0.78 Neuroticism $\alpha$ = 0.84, Extraversion $\alpha$ = 0.76, Openness $\alpha$ = 0.84; Total scale $\alpha$ = 0.88.

*Trait emotional intelligence.* Trait emotional intelligence was measured with the Emotional Development Questionnaire for Primary Education CDE_9-13 [65]. This self-report questionnaire consists of 41 items with an 11-point Likert-type response (0 = *very seldom or never*, 10 = *almost always*), distributed in five dimensions of the pentagonal model of the Psychopedagogical Guidance Research Group of the University of Barcelona (GROP) [68]: emotional awareness, emotional regulation, emotional autonomy, social competence, and life competencies and well-being, as well as a general dimension. In this study, the reliability of the scales measured with Cronbach's $\alpha$ was 0.80, 0.75, 0.62, 0.72, and 0.72, respectively. The reliability of the full scale was 0.91.

### 3. Results

#### 3.1. The Location of Trait EI in the B5 Factor Space

We found moderate to high correlations between trait EI and the B5 (see Table 1). Trait EI correlated the highest with Agreeableness (r = 0.70) and the lowest with Extraversion (r = 40). The mean inter-correlation between the global trait EI and the B5 was r = 0.55.

**Table 1.** Inter-correlations among trait EI, the B5, the B2, and the GFP.

| | GPF | Stability | Plasticity | O | C | E | A | N | Total EI |
|---|---|---|---|---|---|---|---|---|---|
| GPF | 1 | | | | | | | | |
| Stability | 0.783 ** | 1 | | | | | | | |
| Plasticity | 0.423 ** | 0.73 ** | 1 | | | | | | |
| Openness | 0.808 * | 0.650 ** | 0.836 ** | 1 | | | | | |
| Conscientiousness | 0.912 ** | 0.862 ** | 0.807 * | 0.782 ** | 1 | | | | |
| Extraversion | 0.584 ** | 0.568 ** | 0.833 ** | 0.392 ** | 0.563 ** | 1 | | | |
| Agreeableness | 0.851 ** | 0.762 ** | 0.699 ** | 0.625 ** | 0.761 ** | 0.541 ** | 1 | | |
| Neuroticism | −0.430 ** | 0.135 * | −0.261 ** | −0.326 | −0.327 ** | −0.109 ** | −0.361 ** | 1 | |
| Emotional Awareness | 0.591 ** | 0.484 ** | 0.498 ** | 0.436 ** | 0.509 ** | 0.394 ** | 0.633 ** | −0.266 ** | 0.861 ** |
| Emotional Regulation | 0.585 ** | 0.266 ** | 0.406 ** | 0.425 ** | 0.531 ** | 0.252 ** | 0.536 ** | −0.294 ** | 0.756 ** |
| Autonomy | 0.348 ** | 0.334 ** | 0.367 ** | 0.334 ** | 0.394 ** | 0.282 ** | 0.335 ** | −0.257 ** | 0.727 ** |
| Social Competence | 0.543 ** | 0.415 ** | 0.453 ** | 0.417 ** | 0.470 ** | 0.339 ** | 0.622 ** | −0.335 ** | 0.839 ** |
| Life and Wellbeing Competencies | 0.567 ** | 0.375 ** | 0.413 ** | 0.344 ** | 0.511 ** | 0.345 ** | 0.528 ** | −0.398 ** | 0.828 ** |
| Total emotional intelligence (0.89) | −0.693 ** | 0.463 ** | 0.556 ** | 0.491 ** | 0.600 * | 0.397 ** | 0.700 ** | −0.484 ** | 1 |

Note. *n* = 259. GPF = General Personality Factor, O = Openness, C = Conscientiousness, E = Extraversion, A = Agreeableness, N = Neuroticism, Total EI = Emotional Intelligence total coefficient. * $p < 0.05$, ** $p < 0.01$.

Multiple regression analysis with trait EI as the criterion variable and B5 as predictors showed that Agreeableness and Neuroticism significantly predicted trait EI (see Table 2, Section 1), accounting for 57.1% of its variance. All VIF values were below five, showing no problematic multicollinearity issues.

**Table 2.** Multiple Regression of trait EI onto the B5 and trait EI onto the B2.

| Section Model | | Unstandardized Coefficients | | Standardized Coefficients | *t* | *p* |
|---|---|---|---|---|---|---|
| | | B | Std. Error | Beta | | |
| 1 | (Constant) | 2.777 | 0.488 | | 5.686 | 0.000 |
| | Conscientiousness | 0.012 | 0.009 | 0.109 | 1.325 | 0.187 |
| | Openness | 0.007 | 0.014 | 0.034 | 0.507 | 0.613 |
| | Extraversion | 0.009 | 0.011 | 0.040 | 0.777 | 0.438 |
| | Agreeableness | 0.096 | 0.013 | 0.477 | 7.241 | 0.000 |
| | Neuroticism | −0.043 | 0.007 | −0.284 | −6.378 | 0.000 |
| 2 | (Constant) | 1.003 | 0.567 | | 1.768 | 0.078 |
| | Stability | 0.010 | 0.006 | 0.122 | 1.616 | 0.107 |
| | Plasticity | 0.061 | 0.010 | 0.467 | 6.175 | 0.000 |

Note. *n* = 259.

We ran a principal axis factoring exploratory analysis of the 65 BFQ-NA items and the 41 CDE_9-13 items with no rotation. We obtained five factors instead of six and did not find a trait EI factor. Instead, the 41 trait EI items loaded in all five factors mixed with many BFQ-NA items. The results obtained indicated that different dimensions of emotional intelligence loaded in different factors in conjunction with different personality traits (see Table A1 in Appendix A).

### 3.2. Trait EI and the B2

We investigated the relationship of trait EI with the B2. We computed an Alpha/Stability coefficient adding conscientiousness to agreeableness and neuroticism. We also calculated a Beta/Plasticity variable adding Openness to Extraversion. Both coefficients correlated positively (r = 0.73, *p* < 0.01).

As observed in Table 1, the correlation between trait EI and Alpha/Stability (r = 0.46) was lower than that between trait EI and Beta/Plasticity (r = 0.56). The B2 explained a substantial 31% of the variance in Trait EI, with Beta/Plasticity as the strongest and only significant predictor (Table 2, Section 2).

Therefore, these results indicated that all five dimensions of emotional intelligence correlated more strongly with Plasticity than Stability.

### 3.3. Trait EI and the Big One

We retained one unique unrotated general component (GFP) to identify the general personality factor. The correlation between trait EI and the GFP was r = 0.69.

Following Van der Linden et al. [55], we also performed a hierarchical regression analysis including trait EI as the criterion, the General Personality Factor (GFP) as predictor entered in step 1, and the individual B5 scales (O, C, E, A, and N) as predictors entered in step 2. The GFP in step 1 explained a substantial 48% of the variance in trait EI ($R^2$adj:47.5, $F_{(1,116)}$ = 102.08, *p* < 0.001). Concerning the total unique variance of the B5 scale scores in step 2, they explained an additional 7.9% of trait EI variance ($R^2$adj:55.4, $F_{(6,111)}$ = 24.91, *p* < 0.001).

We combined the five BFQ-NA scales with the five CDE_9-13 factors and submitted them to a single-factor PCA using Oblimin (delta = 0) rotation (see Table 3). The analysis of the eigenvalues and the scree plot indicated the existence of only one factor. The loadings of the B5 on this factor ranged from 0.53 to 0.85, while those of trait EI ranged from 0.69 to 0.80. The trait EI dimensions that loaded highest were Agreeableness and Conscientiousness. Still, Emotional Awareness, Emotional Regulation, Social Competence, and Life and Well-Being Competencies loaded higher than Neuroticism, Openness, and Extraversion.

**Table 3.** Principal Component Analysis of trait EI and B5 dimensions.

|  | Component |
|---|---|
|  | 1 |
| Agreeableness | 0.849 |
| Responsibility | 0.808 |
| Emotional Awareness | 0.803 |
| Social Competence | 0.799 |
| Life & Wellbeing Competencies | 0.778 |
| Emotional Regulation | 0.728 |
| Openness | 0.698 |
| Autonomy | 0.688 |
| Extroversion | 0.568 |
| Neuroticism | −0.533 |

Note. *n* = 259.

## 4. Discussion

We had three main objectives in the present study. The first was the analysis of the relationship between trait emotional intelligence (EI) and the Big Five (B5) model of personality. The second objective was to assess the correlation between trait EI and the two Big Five factors (B2). The last one was to test the relationship between trait EI and the Big One personality factor (B1). This study used a sample of Spanish children aged 9 to 13 years and a new measure of trait EI developed by Pérez-Escoda et al. [65].

Regarding the relationship between trait EI and the B5 model of personality, our findings showed that only two of the five personality traits, Agreeableness and Neuroticism, significantly predicted trait EI. This result is different from the obtained in other studies in adults, where all B5 scales significantly predicted trait EI [35,58–60]. There may be different reasons for this discrepancy. On the one hand, it may be due to differences between the personality structures of adults and young people [64]. According to research on children and adolescents, extraversion, agreeableness, conscientiousness, stability, and openness decrease from childhood to adolescence [63,64]. Therefore, it is possible that trait EI behaves as a distinct personality dimension for adults but does not behave in the same way for children. If that were the case, it would mean that adults and children have not only an age difference but also some qualitative differences in personality and emotional development.

On the other hand, we used a different theoretical framework of trait emotional intelligence and a different measurement instrument. We relied on Bisquerra and Pérez-Escoda's [68] theoretical framework and used a measure of trait EI that offers five dimensions: emotional awareness, emotional regulation, social competence, emotional autonomy, and competence for life and well-being [65]. Other research studies were based on Petrides and Furnham's definition of emotional intelligence [43]. They used other measures of trait EI such as the Trait Emotional Intelligence Questionnaire—Child Form (TEIQue-CF) [69], Trait Emotional Intelligence Adolescent Questionnaire (TEIQue-ASF) [70], or the Emotional Intelligence Questionnaire (EIQ) [71], all of which contemplate different components of emotional intelligence. The different measurement tools and trait EI dimensions may explain why the results diverge.

The second objective was to investigate the relationship of trait EI with the B2. The correlations were confirmed. The results indicated that all five dimensions of emotional intelligence correlated more strongly with Plasticity than Stability. This result is different from the one obtained by Alegre et al. [60], in which Emotional Regulation, Life and Well-being competencies, and Autonomy are related more strongly to Stability than to Plasticity. In contrast, Emotional Awareness and Social Competence correlated more strongly to Plasticity.

Our last objective was to test the relationship between trait EI and the Big One personality factor (B1). The correlation between this GFP and trait EI was strong. This correlation was similar to other studies with adults [37,58–60] and children [61]. In our study, the trait

EI dimensions with the highest loadings were Agreeableness and Conscientiousness. In addition, Emotional Awareness, Emotional Regulation, Social Competence, and Life and Well-Being competencies loaded higher than Neuroticism, Openness, and Extraversion. Whereas, in the study of Kamamoto et al. [61], Extraversion, Agreeableness, Conscientiousness, and Openness were positively associated, and Neuroticism was negatively associated with trait EI scores, consistent with the directions of the factor loadings of the GFP. Our research reveals that GFP was significantly associated with trait EI, similarly to other studies with children [61,63]. Therefore, our study reinforces the thesis that trait EI can be considered the same as GFP or, as Rushton and Irwing [34] (p. 146) put it, the culmination of the GFP. Research on trait EI revealed that children with higher scores show more adaptive coping styles and prosocial behaviors [52,72]. Therefore, GPF can be considered an equivalent of trait EI that conceptually overlaps with EI [35,59], likewise as a social efficacy factor [37,73,74]. In addition, a meta-analysis study developed by Roberts et al. [75] points out how personality traits can be modified through intervention. Therefore, it makes sense to develop social-emotional programs in schools at different stages of schooling.

Our study has some limitations that are important to note. First, we used questionnaires to obtain all the data. Also, the same respondents answered all the questionnaires. Monomethod and mono response studies tend to find artificially inflated correlations. In this study, the correlations are robust, and we believe that this eliminates the possibility of false correlations.

## 5. Conclusions

The novelty of our research, with respect to other studies, is that it was conducted with a sample of Spanish children between 9 and 13 years of age and used a different definition and measurement of trait EI known as the Emotional Development Questionnaire for Primary Education (CDE_9-13) [65].

Our results provide some support for the convergent validity of the B5 concerning trait EI. However, they do not support the discriminant validity between the B5 and trait EI. On the other hand, the correlations between trait EI and the B2 were confirmed in our study. Finally, our results confirm that the trait EI and the GFP are the same constructs. Since, by definition, people with high trait EI possess the ability to adapt and thrive in our society, this advantage also applies to the GFP. These conclusions are essential for the field of education as it implies and supports the need to introduce social and emotional development programs from the earliest years of schooling to promote adaptability and social efficacy.

**Author Contributions:** Conceptualization, È.L.-C., N.P.-E. and A.A.; methodology, N.P.-E. and A.A; validation, N.P.-E. and A.A.; investigation, È.L.-C., N.P.-E. and A.A.; data curation, È.L.-C.; writing—original draft preparation, È.L.-C., N.P.-E. and A.A; writing—review and editing, È.L.-C., N.P.-E. and A.A. All authors have read and agreed to the published version of the manuscript.

**Funding:** This research was funded by the Research Groups 2021 Grant from the Faculty of Education of the University of Barcelona.

**Institutional Review Board Statement:** The study was conducted according to the guidelines of the Declaration of Helsinki in 1964 and its later amendments. It followed the recommendations of the Bioethics Commission of the University of Barcelona.

**Informed Consent Statement:** Informed consent was obtained from all subjects involved in the study.

**Data Availability Statement:** Data are available on request due to privacy and ethical restrictions.

**Acknowledgments:** The authors are particularly grateful to all schools, children, and their families for their participation in the study.

**Conflicts of Interest:** The authors declare no conflict of interest.

# Appendix A

**Table A1.** Combined Factor analysis of trait EI and B5 items.

| | Components | | | | |
|---|---|---|---|---|---|
| | **1** | **2** | **3** | **4** | **5** |
| ITEM9 | 0.716 | | | | |
| ITEM3 | 0.659 | | | | |
| BFQ38 | −0.658 | | | | |
| BFQ32 | −0.657 | | | | |
| ITEM41 | 0.653 | | | | |
| BFQ11 | −0.623 | | | | |
| BFQ7 | −0.616 | | | | |
| ITEM32 | 0.613 | | | | |
| BFQ28 | −0.587 | | | | |
| BFQ45 | −0.581 | | | | |
| BFQ34 | −0.578 | | | | |
| ITEM27 | 0.577 | | | | |
| ITEM24 | 0.571 | | | | |
| ITEM10 | 0.563 | | | | |
| BFQ22 | −0.563 | | 0.365 | | |
| BFQ62 | −0.557 | | 0.360 | | −0.334 |
| ITEM38 | 0.556 | | 0.305 | | |
| BFQ52 | −0.556 | | 0.346 | | |
| ITEM37 | 0.550 | | | | |
| ITEM22 | 0.549 | | | | |
| BFQ65 | −0.540 | | | | |
| ITEM25 | 0.536 | | | | 0.319 |
| ITEM6 | 0.534 | | | | |
| ITEM33 | 0.524 | | | 0.348 | |
| BFQ63 | −0.523 | | | 0.399 | |
| BFQ49 | 0.515 | 0.422 | | | |
| ITEM28 | 0.508 | | | 0.343 | |
| ITEM40 | 0.502 | | | | |
| ITEM23 | 0.499 | | | | 0.337 |
| ITEM5 | 498 | | | | |
| BFQ30 | −0.497 | | 0.392 | | −0.375 |
| ITEM1 | 0.494 | | | | |
| BFQ12 | −0.490 | | 0.328 | | −0.347 |
| BFQ48 | −0.487 | | 0.344 | | |
| BFQ3 | −0.487 | | | | |
| ITEM18 | 0.484 | | 0.419 | | |
| BFQ25 | −0.484 | | | | |
| ITEM36 | 0.474 | | | | 0.370 |
| BFQ13 | −0.472 | | | | |
| ITEM12 | 0.472 | | | | |
| BFQ20 | −0.469 | | | | |
| ITEM31 | 0.467 | | | | |
| ITEM8 | 0.464 | | | 0.345 | |
| BFQ51 | −0.457 | | | | |
| BFQ57 | −0.453 | | | | |
| BFQ26 | −0.452 | | | | −0.303 |
| BFQ6 | 0.445 | 0.392 | | | |
| ITEM16 | 0.440 | | 0.319 | | |
| ITEM39 | 0.438 | | 0.315 | | |
| BFQ2 | −0.438 | | | | |
| BFQ5 | −0.437 | | 0.300 | | −0.318 |
| ITEM11 | 0.437 | | | | |
| BFQ53 | −0.429 | | | | |
| BFQ18 | −0.426 | | | | |

**Table A1.** *Cont.*

| | Components | | | | |
| --- | --- | --- | --- | --- | --- |
| | **1** | **2** | **3** | **4** | **5** |
| BFQ16 | −0.421 | | | | |
| BFQ37 | −0.420 | | 0.343 | | |
| ITEM13 | 0.414 | | | | |
| ITEM2 | 0.409 | | 0.369 | | |
| ITEM19 | 0.408 | | 0.384 | −0.354 | |
| ITEM7 | 0.400 | | | | |
| BFQ60 | −0.396 | | | | |
| BFQ24 | −0.378 | | | −0.336 | |
| BFQ41 | 0.376 | 0.358 | | | |
| BFQ27 | −0.376 | | | | |
| BFQ64 | −0.374 | | | | |
| BFQ43 | −0.370 | | | | |
| ITEM20 | 0.350 | | | | |
| ITEM35 | 0.349 | | | | |
| BFQ40 | −0.348 | | | | 0.306 |
| ITEM14 | 0.341 | | | | |
| BFQ21 | −0.327 | | | | |
| ITEM34 | 0.321 | | | | |
| ITEM15 | 0.303 | | | | |
| BFQ39 | | | | | |
| BFQ36 | | | | | |
| BFQ59 | | | | | |
| BFQ33 | | | | | |
| BFQ17 | 0.433 | 0.616 | | | |
| BFQ15 | 0.497 | 0.591 | | | |
| BFQ8 | 0.382 | 0.578 | | | |
| ITEM21 | | 0.567 | | | |
| ITEM29 | 0.390 | 0.540 | | | |
| BFQ4 | | 0.512 | | | |
| ITEM26 | 0.400 | 0.502 | | | |
| ITEM30 | 0.314 | 0.427 | | | |
| BFQ61 | | 0.425 | | | |
| BFQ54 | | 0.417 | | | |
| BFQ29 | 0.306 | 0.396 | | −0.320 | |
| ITEM17 | | 0.387 | | | |
| BFQ9 | | | | | |
| BFQ1 | | | | | |
| BFQ56 | | | | | |
| BFQ46 | −0.345 | | 0.351 | | −0.308 |
| BFQ44 | −0.309 | | 0.326 | | |
| BFQ58 | | 0.387 | | −0.441 | |
| ITEM4 | | | 0.363 | −0.415 | |
| BFQ31 | 0.336 | | | −0.379 | |
| BFQ47 | −0.315 | | | 0.375 | |
| BFQ35 | | 0.311 | | 0.354 | |
| BFQ55 | | 0.310 | | 0.343 | |
| BFQ50 | | | | 0.305 | |
| BFQ42 | | | | | |
| BFQ19 | −0.359 | 0.318 | | | 0.439 |
| BFQ23 | −0.361 | | | | −0.363 |
| BFQ10 | | | | | 0.338 |
| BFQ14 | | | | | |

Extraction Method: Principal Component Analysis.

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
