# Peer review of "The Relationship between Children’s Trait Emotional Intelligence and the Big Five, Big Two and Big One Personality Traits"

_education, doi:10.3390/educsci12070491_

Round 1
Reviewer 1 Report
The article is devoted to an interesting and relevant problem - the study of the relationship between emotional intelligence and personality traits. The article acquires special significance due to the need to develop the emotional intelligence of children for the development of personality in the educational environment, thereby expanding the subject field of child psychology and personality psychology.
However, the content of the article raises several methodological remarks:
- It is necessary to edit the abstract: focus on the specific results of the study obtained by the authors (Confirmation of the results obtained in the studies of other scientists is wonderful, but it is important for the article - what is the novelty, value and originality of the results of this study).
- The authors provide an overview of current research on personality, Five, Big Two and Big One personality traits, which does not end with the formulation of the authors' theoretical position. In this connection, in the Introduction, the authors need to add: a theoretical generalization based on the results of the analysis, outlining their scientific position, to formulate research hypotheses.
- In conclusion, the authors need to formulate generalized conclusions based on the results of the study, and not repeat the results of the study.
Reviewer 2 Report
This paper presents original data checking associations between trait EI and personality (FFM) among children. As far as I know, just one study, which is not cited in the current research, examines this issue among children and adolescents (Kawamoto et al., 2021).
Despite its originality, the study should reconsider its theorical framework in-depth. There are studies that should be added or examined in more detail in the manuscript (e.g., Tackett et al., 2012; Van de Linden et al., 2017; Van der Linden et al., 2018). Instead of that, the authors lead their findings to their own previous research repeatedly.
The study posits contradictory views in the relationships between trait EI and GFP across the paper. Despite that many studies have proven that both variables are virtually the same (Van de Linden et al., 2017; Van der Linden et al., 2018), they introduce the concepts as different constructs to suggest that trait EI is a proxy for GFP just at the conclusions.
Additionally, I really encourage the authors to search the explanation in the discrepancies observed between adults and children according to the relationship trait EI-personality in the course of personality development.
Furthermore, in the conclusions, the applicability of the results is not coherent, and requires much more theoretical and empirical support.
Here below I suggest some changes in the manuscript by sections.
Introduction
L: 27-30. “… that the benefits of developing socio-emotional skills in children go beyond cognitive development and academic results; additionally, they are essential drivers of health, academic performance, and labor market prospects”. Remove “academic performance” because the term is conceptually the same that “academic results”, previously aforementioned.
L: 37-40. “Hence, emotional education is one of the life skills that should be taught”. It is more correct to use another verb like “emphasized” instead of “taught”. You should remove the last sentence in the paragraph: “So being emotionally intelligent would presumably be a good help in achieving this” because it is tautologous, repeating the same idea twice in the paragraph.
A deeper theorical explanation in the links between socio-emotional skills, emotional education and EI is required when EI is introduced in the text, as well as a more parsimonious and structural content. For instance: 1) Importance of emotional education in current society, 2) Positive outcomes of socio-emotional skills based on emotional education, 3) The role of EI, explaining its relevance on the development of socio-emotional skills; and 4) Association between EI and personality.
L: 67-70. I highly recommend discussing the findings from Tackett et al. (2012).
The purpose of the study is not clear. Do you want to test the link between trait EI and personality in children to compare the findings to those found among adults?
There are studies that you need to present in the introduction strongly (Kawamoto et al., 2021, which has a similar study purpose; and the meta-analyses conducted by Van der Linden and colleagues; 2017 and 2018).
Materials and methods
Mathematic symbols must be written in italics (e.g., SD) as well as anchors of the scales (e.g., "Completely false for me").
You should specify the Crobach’s alphas of each FFM factor obtained in your study.
Results
Comparisons between your results and those from other studies should be relocated in the Discussion.
Discussion
L: 350. Do not use the verb “survive”.
Add findings from Tacket et al. (2012), Soto et al. (2011) and Soto (2016) to posit alternative explanations in the differences in the associations between FFM and trait EI between adults and children based on personality development course.
L: 353-354: “This conclusion is essential for the field of education as it implies and supports the need to introduce social and emotional development programs from the earliest years of schooling”. Why? You should support your idea deeper or remove it.
References
Some references do not indicate the last number of pages.
I strongly recommend the review of the following studies for their furhter inclusion (or examination in more detail in the paper):
Kawamoto, T., Kubota, A. K., Sakakibara, R., Muto, S., Tonegawa, A., Komatsu, S., & Endo, T. (2021). The General Factor of Personality (GFP), trait emotional intelligence, and problem behaviors in Japanese teens. Personality and Individual Differences, 171, 110480.
Roberts, B. W., Luo, J., Briley, D. A., Chow, P. I., Su, R., & Hill, P. L. (2017). A systematic review of personality trait change through intervention. Psychological Bulletin, 143(2), 117-141.
Soto, C. J. (2016). The little six personality dimensions from early childhood to early adulthood: Mean‐level age and gender differences in parents' reports. Journal of Personality, 84, 409-422.
Soto, C. J., John, O. P., Gosling, S. D., & Potter, J. (2011). Age differences in personality traits from 10 to 65: Big Five domains and facets in a large cross-sectional sample. Journal of personality and social psychology, 100(2), 330-348.
Tackett, J. L., Slobodskaya, H. R., Mar, R. A., Deal, J., Halverson Jr, C. F., Baker, S. R., ... & Besevegis, E. (2012). The hierarchical structure of childhood personality in five countries: Continuity from early childhood to early adolescence. Journal of Personality, 80(4), 847-879.
Van der Linden, D., Pekaar, K. A., Bakker, A. B., Schermer, J. A., Vernon, P. A., Dunkel, C. S., & Petrides, K. V. (2017). Overlap between the general factor of personality and emotional intelligence: A meta-analysis. Psychological bulletin, 143(1), 36-52.
Van der Linden, D., Schermer, J. A., de Zeeuw, E., Dunkel, C. S., Pekaar, K. A., Bakker, A. B., ... & Petrides, K. V. (2018). Overlap between the general factor of personality and trait emotional intelligence: a genetic correlation study. Behavior Genetics, 48(2), 147-154.
Round 2
Reviewer 2 Report
Congratulations. The manuscript has been improved and now is much more clear and presents a proper theoretical background.